# The Effect of Furrow Opener and Disc Coulter Configurations on Seeding Performance under Different Residue Cover Densities

Davut Karayel [1,2,*], Eglė Jotautienė [2] and Egidijus Šarauskis [2]

1   Department of Agricultural Machinery and Technologies Engineering, Faculty of Agriculture, Akdeniz University, 07070 Antalya, Turkey
2   Department of Agricultural Engineering and Safety, Faculty of Engineering, Agriculture Academy, Vytautas Magnus University, Studentu Str. 15A, LT-52262 Kaunas, Lithuania; egle.jotautiene@vdu.lt (E.J.); egidijus.sarauskis@vdu.lt (E.Š.)
*   Correspondence: dkarayel@akdeniz.edu.tr

**Abstract:** The performance of the no-till seeder is one of the most important factors that affect the success of the no-tillage. Striking the right balance between furrow opener design and residue cover is essential for optimizing seeding conditions and ensuring sustainable agricultural practices that promote both soil conservation and high-yield crop production. This study investigates the impact of residue cover on no-tillage maize seeding after wheat harvest, focusing on plant spacing, seeding depth, mean emergence time, and percent emergence. Trials with hoe-type and double-disc-type furrow openers, accompanied by plain- or ripple-disc-type coulters, were conducted in Antalya, Turkey. The results indicate that residue cover had no significant effect on mean plant spacing, but a higher residue cover increased spacing variation. The seeding depth in hoe-type furrow opener trials remained consistent, while double-disc-type furrow openers showed lower depths with 80% and 90% residue covers. The percentage of plant emergence and mean emergence time decreased as the residue cover increased in double-disc-type furrow opener trials. At 90% residue cover, PE decreased to 60%. The impact of disc coulters on hoe-type furrow openers was limited, but they increased seeding depth and MET in double-disc-type furrow openers. These findings can help optimize residue management for improved efficiency in no-till farming systems.

**Keywords:** no-tillage; seeder; coulter; seeding depth; maize; percent emergence

## 1. Introduction

The adoption of no-till farming practices has become increasingly widespread due to its numerous environmental and economic benefits. However, successful implementation hinges on effectively managing crop residues left on the field surface after harvest.

The furrow opener–residue cover relation plays a crucial role in modern agricultural practices, particularly in conservation tillage systems. Herein lies the critical importance of no-tillage furrow opener performance. The amount and type of residue, influenced by the harvested crop, harvest method, and climatic conditions, significantly impact the performance of furrow openers in no-till seeders. These specialized tools cut through and manage residue while creating furrows for seed placement, directly impacting crop establishment success. Optimizing their design and operation across varying residue densities is crucial for ensuring uniform seed depth and good seed–soil contact, and, ultimately, maximizing yield potential in conservation agriculture systems. By understanding how different furrow openers interact with varying residue levels, more effective no-tillage strategies can be developed, promoting sustainable agricultural practices and ensuring food security for the future. Efficient residue management helps increase the performance of the no-till seeder and, therefore, the success of the no-tillage [1].

Previous research has highlighted the crucial role of furrow openers in creating optimal seedbeds for germination and establishment. Studies such as Köller [2], Ahmad et al. [3],

and McLaughlin et al. [4] emphasize the importance of residue management for maximizing no-tillage success. They point out that furrow openers must effectively penetrate the soil while minimizing surface residue disturbance. An effective and efficient furrow opener should have two primary features: precise soil management and minimal demands on draft and vertical force. When integrated into a no-tillage seeding system, furrow openers should also support the functions of seeding system components, such as maintaining an adequate surface residue distribution, accurately and uniformly placing seeds and fertilizer, and ensuring regular inter-plant spacing [5–7].

Several design features contribute to a furrow opener's performance in residue-laden conditions. Celik [8] and Rouzbeh [9] describe typical no-till seeders comprising chisel irons, furrow openers, closing wheels, and seed units. Each component plays a specific role: chisel irons mark and disturb the surface, furrow openers create seed furrows, closing wheels ensure seed–soil contact, and seed units deliver seeds efficiently. Raoufat and Matbooei [10] investigated the efficacy of furrow cleaners under varying residue densities and travel speeds in maize–wheat rotations. Their findings suggest that while higher speeds (10 km/h) enhance cleaner performance, the optimal seed distribution occurs at moderate speeds (7 km/h). Furthermore, they demonstrated the ability of furrow cleaners to significantly reduce in-furrow residue accumulation (45–70%) [10]. Zeng and Chen [11] conducted a study to compare the performance of five vertical tillage tools in a maize stubble field. The five tools included two coulters with 8 and 13 flutes and three rippled discs with diameters of 457 mm, 508 mm, and 559 mm. All the tools were tested at a working speed of 16 km/h and a tillage depth of 100 mm. The results of this study showed that increasing the diameter of the disc from 457 mm to 559 mm significantly increased the soil disturbance area (by 127%), residue incorporation (by 44%), lateral cutting force (by 79%), and soil opening width (by 30%), but decreased residue cover (by 5%) [11]. In summary, the fluted coulters exhibited greater soil disturbance and residue incorporation compared to the rippled discs, along with an increased draft force and soil adhesion. Šarauskis et al. [12] conducted a study on the theoretical aspects of disc coulters' working process under no-tillage conditions. They discovered that increasing the sharpening angle of the disc coulter blade by one degree can reduce the cutting force required to cut wheat straw by 6.5 N. Furthermore, they found that decreasing the thickness of the disc coulter blade by one millimeter can reduce the cutting force required by 12.5 N [12].

Understanding the effects of tillage practices on residue distribution further informs no-till management strategies. For this purpose, Schneider et al. [13] compared the impacts of plowing, rotary tillage, and chisel plowing on rapeseed stubble distribution. They observed that plowing buries nearly all residues, while rotary tillage leaves the majority (70%) on the surface. Interestingly, chisel plowing exhibited better residue incorporation than discing. Zhou et al. [14] conducted an experiment to compare the effectiveness of three different tillage tools for straw incorporation in rice stubble fields. The tools evaluated included a conventional rotary tiller, a straw rotary burying and returning implement, and a combination of subsoiling with straw rotary burying and returning equipment. The findings indicated that both the straw rotary burying and returning equipment and the subsoiling combined with straw rotary burying and returning equipment (SSR) were notably effective in cutting and integrating straw into the soil. Additionally, the SSR was able to bury more straw into the lower layer of soil. Therefore, the SSR was recommended as the most effective tillage tool for fields with excessive residue concerns [14].

A study conducted by Jiang et al. has proposed some future development trends for minimum or no-till seeders [15]. These trends include strengthening research on basic theories and integration mechanisms, building a data-sharing platform for seeding operations, and establishing or improving specific systems to enhance the performance of minimum and no-tillage seeders. As part of the third development trend, this research aims to enhance the efficacy of the soil-engaging components of seeding machinery. To achieve this, an investigation was conducted on the efficiency of different configurations of furrow openers and disc coulters, when dealing with varying levels of residue cover densities

after wheat harvest. This study mainly focuses on evaluating how different surface residue levels impact the performance of furrow openers and, consequently, seeding success. By understanding these interactions, furrow opener design and selection can be optimized for improved residue management and enhanced performance.

## 2. Materials and Methods

The seeder utilized in the trials was a vacuum-type precision seeder (Figure 1). The seed metering mechanism of the seeder derives its movement from the carrying wheels. The transmission system underwent adjustments to facilitate seeding at a distance of 203 mm between seeds. The theoretical seeding depth of the seeder was modifiable by adjusting the vertical level between the press wheel and the furrow opener. Throughout the trials, the seeder was configured to plant at a theoretical depth of 50 mm. Plain or ripple disc coulters were positioned in front of the furrow openers to cut stubble and loosen the soil.

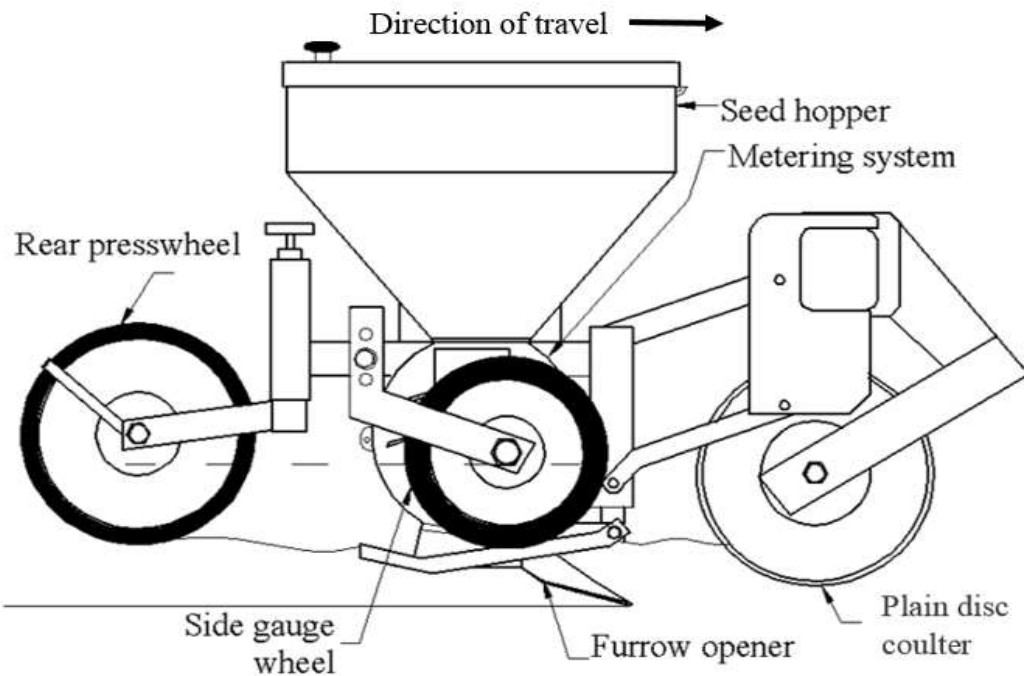

**Figure 1.** The seeder and main components.

The hoe-type furrow opener employed in the trials comprised an 8 mm thick cast-end iron and two 4 mm thick wings. Seeds descended into the furrow created by the tip iron through the opening formed by the wings. The wings served to keep the line opened by the tip bar from closing until the seed had settled into the furrow, as depicted in Figure 2a.

The double-disc furrow opener was formed from two flat discs supported by 300 mm diameter bearings. At the point where the discs made contact, the angle measured 12°, as illustrated in Figure 2b. This configuration provided an effective mechanism for precise furrow creation and seed placement during the trials.

Furrow coverers, situated after the furrow openers, facilitated the coverage of seeds with loose soil. These coverers were adjustable based on spring pressure to ensure optimal coverage, taking into account the varying tempering conditions of the soil.

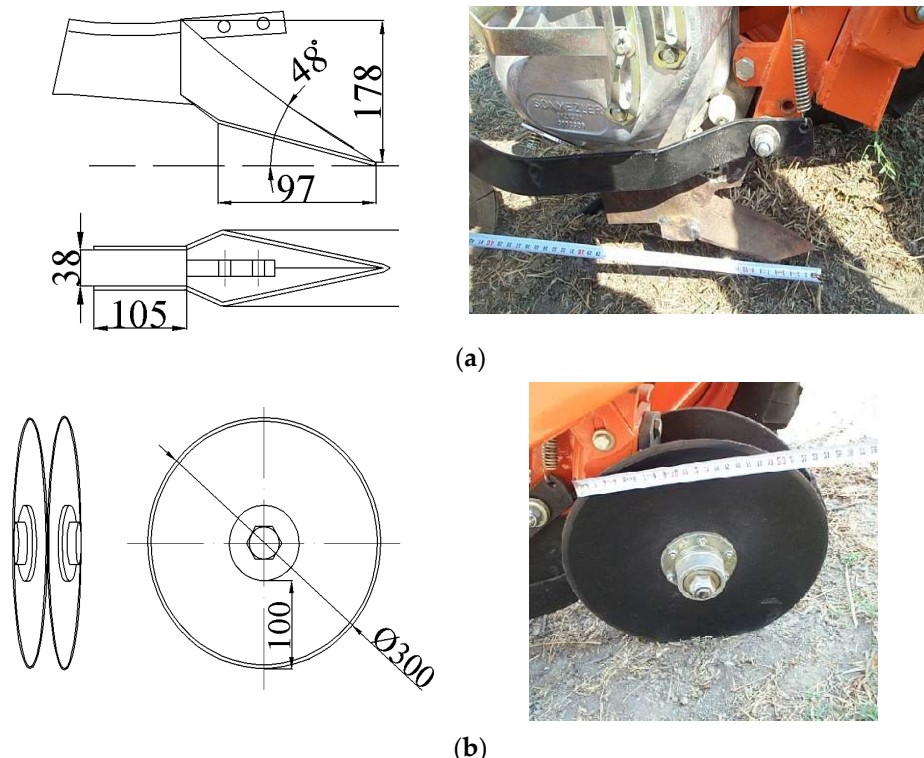

**Figure 2.** Technical drawings and photos of furrow openers. (**a**) Hoe-type furrow opener, (**b**) double-disc-type furrow opener.

The maize seeds utilized in the experiments exhibited specific characteristics: a thousand-grain weight of $238 \pm 2.1$ g, a sphericity of $77 \pm 0.6\%$, a laboratory emergence rate of $98 \pm 2\%$, and average dimensions of $10.8 \pm 0.2$ mm in length, $5.3 \pm 0.05$ mm in thickness, and $7.7 \pm 0.11$ mm in width. The experimental study was conducted in a field in Antalya province, Turkey, whose soil is classified as silty loam, consisting of 37.1% sand, 26.1% silt, and 36.8% clay. The average percentage moisture content of the 0–20 cm top layer of the soil just before the seeding was measured at $24.3 \pm 5.3\%$.

Then, 400 mm diameter plain or ripple disc coulters were mounted in front of furrow openers. The ripple disc coulter had 13 ripples. The spacing from the center of every wavy-edged disc to the front of the furrow opener was established at 450 mm (Figure 3).

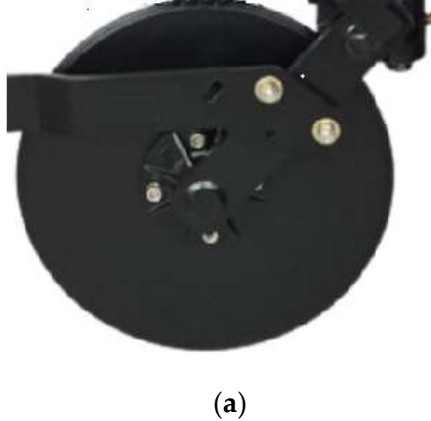

(**a**)

**Figure 3.** *Cont.*

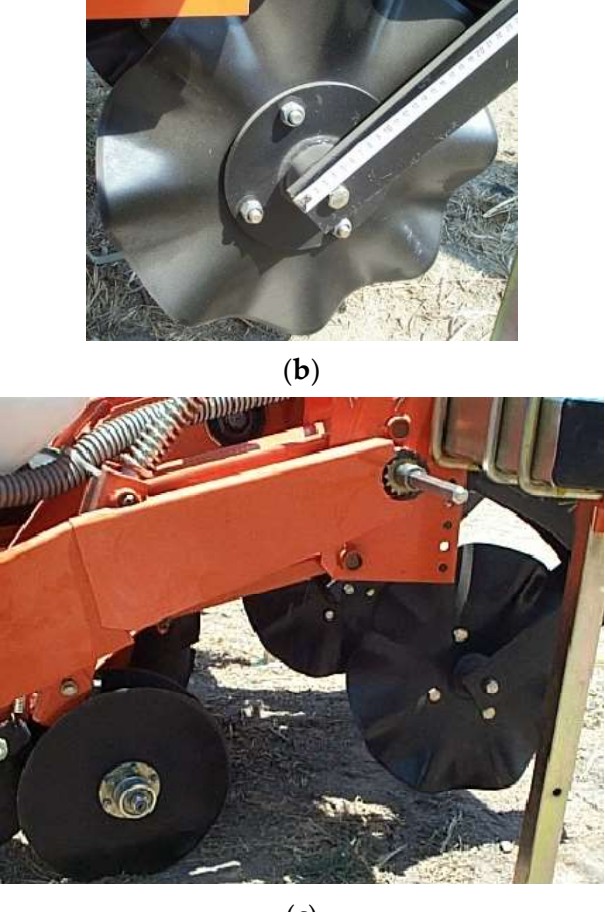

(**b**)

(**c**)

**Figure 3.** Disc coulters used in the experiments. (**a**) Plain disc, (**b**) ripple disc, (**c**) position of the disc coulters in front of the furrow openers.

After the seeding operation, an evaluation was carried out to assess seed distribution along the row length, the consistency of seeding depth, mean emergence time (MET), and percent emergence (PE). Field measurements were carried out 17 days post seeding, encompassing approximately 50 maize plants for each treatment, to determine plant spacing and seeding depth. The distances between consecutive plants within each furrow were quantified, and subsequent computations yielded the mean plant spacing and the coefficient of variation in spacing based on these field measurements. Furthermore, the depths of seeds beneath the soil surface were determined by marking the plant at ground level, subsequently extracting the entire stem length below the mark to establish the effective seeding depth. The mean seeding depth and the coefficient of variation in depth were derived from these depth measurements. The coefficient of variation in spacing or depth was calculated using the following equation [10].

$$SD = \sqrt{\frac{\sum_{i=1}^{N}(x_i - \overline{x})^2}{N-1}} \tag{1}$$

$$CV = \frac{SD}{\overline{x}} \tag{2}$$

where $SD$ is the standard deviation, $\overline{x}$ is the theoretical seed spacing or seeding depth, $x_i$ is the measured seed spacing or seeding depth, $N$ is the total number of measurements, and $CV$ is the coefficient of variation.

Furthermore, daily seedling counts were performed throughout the emergence period within a 25 m row segment for each treatment. Based on these counts, *MET* and *PE* were computed, following the methodology described by Bilbro and Wanjura, and Chen et al. [16,17].

$$PE = \frac{N_{te}}{n} \times 100 \tag{3}$$

$$MET = \frac{N_1 D_1 + N_2 D_2 + \ldots N_n D_n}{N_1 + N_2 + \ldots N_n} \tag{4}$$

where *PE* is the percent emergence, *MET* is the mean emergence time, $N_{te}$ is number of emerged seedlings per meter, *n* is initial number of seeds sown per meter, $N_{1\ldots n}$ is the number of new seedlings emerging on each successive day, and $D_{1\ldots n}$ is number of days elapsed after the seeding operation.

The investigation employed a split-split plot design with three replications. The experimental design and test parameters are presented in Table 1. Randomization techniques were employed during the experiment's setup to ensure the spatial distribution uniformity of variable factor selection. Each treatment combination of furrow opener and disc coulter configurations was randomly assigned within each replicate block to mitigate potential biases arising from spatial effects. These measures were implemented to enhance the validity and reliability of our experimental results by minimizing the influence of spatial variability on the observed outcomes. Data were analyzed using SPSS 17 software (SPSS Inc., Chicago, IL, USA). Analysis of variance (ANOVA) was conducted to assess the influence of plant spacing, seeding depth, emergence time, and percent emergence on the measured variables. Duncan's multiple range test was utilized to differentiate significant mean variations. Prior to analysis, data were assessed for normality and homoscedasticity assumptions. When violations were detected, appropriate transformations were applied to ensure normality. Statistical significance was determined at $\alpha = 0.05$.

Conservation tillage is a soil management technique that aims to maintain a minimum of 30% soil surface coverage with crop residue. Estimating residue cover solely by observing across a field, typically done from the road or field edge, tends to overstate the actual coverage. To achieve accurate measurements, researchers must take readings directly within the field, looking down at the soil–residue interface. Several methods, including line-transect, optical, and remote sensing, are used to measure crop residue cover [18].

Papendick [19] and Raoufat and Matbooei [10] introduced Equation (3) to convert the flat surface cover to a specific residue quantity. This equation was used to calculate the residue cover rate in this study.

$$Y = \left(1 - e^{-0.000644X}\right) \times 100 \tag{5}$$

where *Y* is the percent residue cover and *X* is the dried weight of residue per unit surface area, lb acre$^{-1}$.

If the above equation is solved according to a 30% residue cover rate, there must be at least 621 kg/ha of residue on the field surface. The average residue amounts of the plots on which this research was conducted, with different residue cover densities, are 910, 1400, 2850, and 5020 kg/ha. Using the above equation, the approximate percentage rates of residue in these plots were calculated as 40%, 55%, 80%, and 90%, respectively. Therefore, in this study, the effects of percent residue covers of 40%, 55%, 80%, and 90% on the seeding quality for different furrow openers (hoe and double disc types) and disc coulter (control, plain disc, and ripple disc) combinations were examined. The residue rates tested were over the minimum ratio for conservation tillage, 30%.

**Table 1.** Experimental design layout: test parameters.

| Main plot (Factor A) | Level A1: Hoe-type furrow opener | | |
|---|---|---|---|
| Subplot (Factor B) | Level B1: Control (without coulter for residue cutting) | Level B2: Plain-disc-type coulter | Level B3: Ripple-disc-type coulter |
| Sub-subplot (Factor C) | Level C1: 40% residue cover<br>Level C2: 55% residue cover<br>Level C2: 80% residue cover<br>Level C2: 90% residue cover | Level C1: 40% residue cover<br>Level C2: 55% residue cover<br>Level C2: 80% residue cover<br>Level C2: 90% residue cover | Level C1: 40% residue cover<br>Level C2: 55% residue cover<br>Level C2: 80% residue cover<br>Level C2: 90% residue cover |
| Main plot (Factor A) | Level A2: Double-disc-type furrow opener | | |
| Subplot (Factor B) | Level B1: Control (without any coulter for residue cutting) | Level B2: Plain disc coulter | Level B3: Ripple disc coulter |
| Sub-subplot (Factor C) | Level C1: 40% residue cover<br>Level C2: 55% residue cover<br>Level C2: 80% residue cover<br>Level C2: 90% residue cover | Level C1: 40% residue cover<br>Level C2: 55% residue cover<br>Level C2: 80% residue cover<br>Level C2: 90% residue cover | Level C1: 40% residue cover<br>Level C2: 55% residue cover<br>Level C2: 80% residue cover<br>Level C2: 90% residue cover |

To obtain different residue cover densities on the plots, the plots with 40% and 80% residue cover rates were sown at a 20 kg/da seeding rate and harvested at residue heights of 10–15 cm and 20–25 cm, respectively. The plots with 55% and 90% residue cover rates were sown at a 25 kg/da seeding rate and harvested at residue heights of 10–15 cm and 20–25 cm, respectively.

## 3. Results and Discussion

### 3.1. Plant (Seed) Distribution Uniformity in the Row

Table 2 presents the mean plant spacings within the row and their corresponding coefficients of variation, measured at various residue cover percentages. Across different furrow opener and disc coulter combinations, the experiments found no statistically significant difference in mean plant spacing. This indicates that residue cover density had no significant impact on average plant (seed) spacing in the row.

**Table 2.** Mean plant spacing and coefficient of variation (CV) for plant (seed) distribution uniformity in the row.

| Residue Cover Density/Rate (%) | Mean Plant Spacing (mm)/CV of Plant Spacing (%) | | | |
|---|---|---|---|---|
| | Control Plots (without Any Disc Coulter) | Plain-Disc-Type Coulter | Ripple-Disc-Type Coulter | |
| | Hoe-type furrow opener | | | Significance |
| 40 | 205/18.2 | 207/16.6 | 203/17.8 | ns [z] |
| 55 | 206/19.9 | 209/18.7 | 208/18.5 | ns |
| 80 | 210/21.5 | 210/20.8 | 211/21.3 | ns |
| 90 | 212/23.9 | 211/21.2 | 209/23.4 | ns |
| Significance | ns [y] | ns | ns | |
| | Double-disc-type furrow opener | | | |
| 40 | 208/18.8 | 208/16.5 | 207/17.2 | ns |
| 55 | 207/19.1 | 206/17.1 | 205/17.8 | ns |
| 80 | 208/22.0 | 205/20.8 | 204/21.1 | ns |
| 90 | 211/24.3 | 209/21.0 | 212/22.8 | ns |
| Significance | ns | ns | ns | |

[y] Statistically nonsignificant differences within a column at a significance level of $p > 0.05$. [z] Statistically nonsignificant differences within a row at a significance level of $p > 0.05$.

However, examining the plant spacing variation coefficients, the lowest values at the lowest residue cover percentage (40%) were observed. Conversely, increasing the residue cover resulted in a higher plant spacing variation coefficient. This aligns with Aikins et al. [5] and Porichha et al. [20], who found that higher residue density encourages residues to enter furrows during seeding, raising the potential for seed contact. While the mean spacing remained statistically similar, the increased variation coefficient at higher residue densities suggests the potential clumping or under-dispersion of seeds. This warrants further investigation into spatial distribution patterns to assess potential impacts on crop emergence and yield. In our study, high residue density caused residues to enter the furrows, where they subsequently impeded seed placement through rolling and bouncing. These findings are consistent with those reported by Choudhary and Baker [21] and Parihar et al. [22]. In addition to residue cover density, further investigation into soil moisture levels, compaction, and seed characteristics could provide deeper insights into factors influencing plant distribution uniformity.

### 3.2. Seeding Depth Uniformity

The mean seeding depth and coefficient of variation values resulting from the trials are outlined in Table 3. In trials utilizing the hoe-type furrow opener, neither the stubble ratio nor disc coulter application demonstrated statistically significant effects on the mean seeding depth. Conversely, in trials conducted with double-disc-type furrow openers, it was observed that seeding depth was lower in plots with 80% and 90% residue cover ratios compared to those with 40% and 55% residue cover ratios. Furthermore, the implementation of plain- or ripple-disc-type coulter applications in front of the furrow opener resulted in soil and residue cutting, loosening the soil just before the furrow opener, thereby causing an increase in seeding depth. Nevertheless, in plots with 80% and 90% stubble ratios, the seeding depth remained within the range of 29–34 mm. Consequently, it is recommended to incorporate components such as a row cleaner in front of the double-disc-type furrow opener to achieve the theoretical planting depth of 50 mm in high residue cover densities.

**Table 3.** Mean seeding depth and coefficient of variation (CV) for seeding depth uniformity.

| Residue Cover Density/Rate (%) | Mean Seeding Depth (mm)/CV of Seeding Depth (%) | | | |
| --- | --- | --- | --- | --- |
| | Control Plots (without Any Disc Coulter) | Plain-Disc-Type Coulter | Ripple-Disc-Type Coulter | |
| Hoe-type furrow opener | | | | Significance |
| 40 | 50/15.6 | 49/16.1 | 52/16.5 | ns [z] |
| 55 | 52/16.8 | 50/18.8 | 52/16.9 | ns |
| 80 | 52/17.1 | 45/19.2 | 51/18.7 | ns |
| 90 | 48/19.8 | 48/20.1 | 48/19.3 | ns |
| Significance | ns [y] | ns | ns | |
| Double-disc-type furrow opener | | | | |
| 40 | 45Aa [t]/8.8 | 47Ba/8.1 | 48Ba/7.9 | ** |
| 55 | 38Aa/9.3 | 43Ba/8.9 | 43Ba/8.8 | ** |
| 80 | 30Ab/22.5 | 33Bb/18.1 | 34Bb/17.9 | ** |
| 90 | 29Ab/29.5 | 32Bb/23.6 | 31Bb/23.5 | ** |
| Significance | * | * | * | |

[y] Statistically nonsignificant differences within a column at a significance level of $p > 0.05$. [z] Statistically nonsignificant differences within a row at a significance level of $p > 0.05$. [t] Statistically significant differences within a row are denoted by distinct uppercase letters at a significance level of $p < 0.05$. Similarly, different lowercase letters at a significance level of $p < 0.05$ represent significant differences within a column. * Statistically significant differences within a column at a significance level of $p < 0.05$ ** Statistically significant differences within a row at a significance level of $p < 0.05$.

Regarding seeding depth, the primary reason for the difference between the hoe-type furrow opener and the double-disc-type furrow opener lies in the direction of the perpendicular component of the soil resistance force when the furrow opener cuts the

soil. As illustrated in Figure 4, the perpendicular component (FP) of the resultant soil resistance acting on the hoe-type furrow opener is downward, facilitating the furrow opener's penetration into the soil. In contrast, for double-disc-type furrow openers, FP is upward, compelling the disc to emerge from the soil. Consequently, the hoe-type furrow opener is not influenced by residue cover density or disc coulter applications in terms of seeding depth, due to its construction. These results align with the theoretical principles presented by Šarauskis et al. [12], Šarauskis and Vaitauskienė [23], and Liu et al. [24] regarding the performance of furrow openers under no-till conditions.

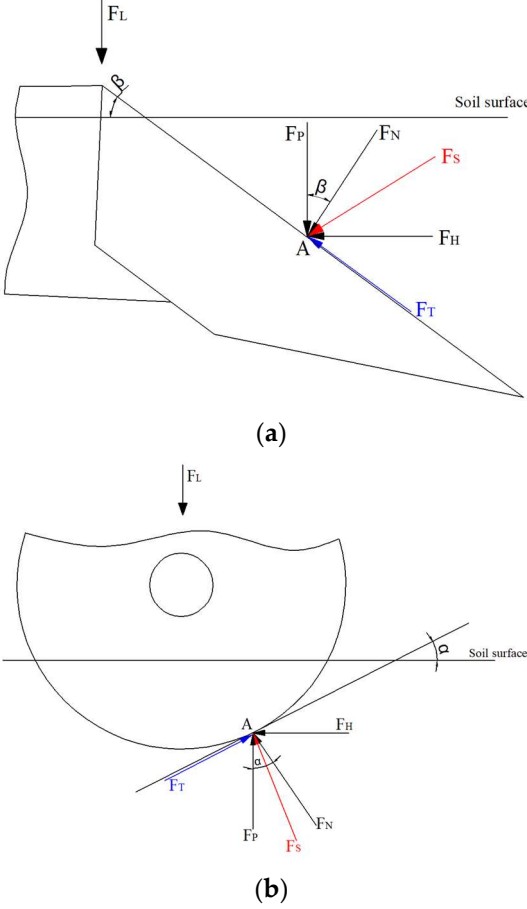

**Figure 4.** Forces acting on (**a**) the hoe- and (**b**) the double-disc-type furrow openers: FL—surcharge load; FN—normal force; FT—frictional force; FH—horizontal force; FP—perpendicular force; FS—soil cutting force (resulting force of soil resistance); A—point of action.

Upon examining the coefficients of variation for seeding depth, it was determined that although the coefficient of variation for furrow openers increased with the rise in residue cover rate, this increase was more pronounced in trials using double-disc-type furrow openers. While both plain and ripple disc applications did not significantly impact the hoe-type furrow opener, the double-disc-type furrow opener reduced the variation in seeding depth and ensured a more uniform seeding depth. This effect is particularly evident at residue cover rates of 80% and 90%. Coupled with the observed effects of residue cover densities, examining the interaction between seeding depth uniformity and soil properties like texture and compaction could provide a comprehensive understanding. Further research endeavors could also delve into the optimization of disc coulter configurations to mitigate the variability in seeding depths, particularly in scenarios of high residue cover, thus enhancing overall seeding performance.

### 3.3. Percent Emergence and Mean Emergence Times

Table 4 presents the replication averages for MET and PE in plots with varying residue cover rates. Statistical analysis revealed that the residue cover rate did not have a significant effect on MET when a hoe-type furrow opener was employed. However, it was observed that MET was higher in plots with 40% and 55% residue cover ratios compared to plots with 80% and 90% stubble ratios in trials utilizing the double-disc-type furrow openers. Results aligning with the seeding depth distribution are shown in Table 3, indicating that the lower seeding depth associated with high residue cover rates, such as 80% and 90% in the double-disc furrow opener, also contributed to a reduced MET value.

**Table 4.** Percent emergence (PE) and mean emergence time (MET).

| Residue Cover Density/Rate (%) | PE (%)/MET (day) | | | |
|---|---|---|---|---|
| | Control Plots (without Any Disc Coulter) | Plain-Disc-Type Coulter | Ripple-Disc-Type Coulter | |
| | Hoe-type furrow opener | | | Significance |
| 40 | 67/9.0 | 68/8.8 | 68/9.1 | ns [z] |
| 55 | 65/8.5 | 66/8.7 | 65/8.7 | ns |
| 80 | 67/8.8 | 69/8.3 | 67/8.8 | ns |
| 90 | 65/8.7 | 63/8.8 | 66/8.8 | ns |
| Significance | ns [y] | ns | ns | |
| | Double-disc-type furrow opener | | | |
| 40 | 70 Aa [t]/7.7Aa | 77Ba/8.2Ba | 81Ba/8.4Ba | ** |
| 55 | 68Aa/7.5Aa | 77Ba/8.1Ba | 80Ba/8.2Ba | ** |
| 80 | 63Ab/6.9Ab | 72Bb/7.5Bb | 73Bb/7.7Bb | ** |
| 90 | 60Ab/6.8Ab | 70Bb/7.5Bb | 64Bc/7.6Bb | ** |
| Significance | * | * | * | |

[y] Statistically nonsignificant differences within a column at a significance level of $p > 0.05$. [z] Statistically non-significant differences within a row at a significance level of $p > 0.05$. [t] Statistically significant differences within a row are denoted by distinct uppercase letters at a significance level of $p < 0.05$. Similarly, different lowercase letters at a significance level of $p < 0.05$ represent significant differences within a column. * Statistically significant differences within a column at a significance level of $p < 0.05$. ** Statistically significant differences within a row at a significance level of $p < 0.05$.

The PE was not statistically affected by the residue cover rate in trials using the hoe-type furrow opener. However, in experiments with the double-disc-type furrow opener, an increase in the residue cover rate negatively impacted the PE. Specifically, in trials with a residue cover rate of 90%, the PE decreased to 60%. This finding is consistent with the observation made by Porichha et al. [20] who reported that a high residue cover rate hinders seed germination by preventing stubble from entering the furrows and making contact with the seeds. Consequently, seeds are deprived of moisture in the soil, impeding their germination.

The influence of disc coulters on PE and MET in the hoe-type furrow opener was found to be statistically insignificant. However, the use of plain- or ripple-type disc coulters was associated with increased MET and PE values. In the case of the double-disc-type furrow opener, the use of cutting discs led to an increase in seeding depth, resulting in elevated MET values.

In the trials conducted using the double-disc-type furrow opener on plots with 80% and 90% stubble ratios, a reduction in seeding depth was observed, leading to a corresponding decrease in the mean emergence time (MET). However, it is noteworthy that, as highlighted by Sen et al. [25] and Masilamani et al. [26], an excessively low seeding depth poses a risk to seed viability, particularly in regions with high temperatures, where the upper soil layer tends to dry rapidly, limiting the seeds' access to moisture. Consequently, the diminished seeding depth associated with 80% and 90% stubble ratios not only impacted the ME but also resulted in a decline in PE. During the trials, it was noted that seeds came into contact with stubble in plots featuring a high residue cover rate (80–90%)

when utilizing the double-disc-type furrow opener. This contact hindered the seeds' ability to absorb soil moisture. Additionally, trials with hoe-type furrow openers experienced intermittent blockages due to the elevated residue cover rate.

Comparing the PE values of hoe-type and single-disc-type furrow openers reveals a lower PE in these plots, even though hoe-type furrow openers achieved the desired theoretical seeding depth. According to Choudhary and Baker [21], Parihar et al. [22], and Karayel and Ozmerzi [27], the hoe-type furrow opener, while cutting the soil, loosens it and deposits seeds into this loosened soil layer. However, this loose layer may quickly dry out, impeding seed germination. In contrast, furrow openers with wider sinking angles, such as double-disc-type furrow openers, compact the soil at the furrow's bottom and deposit seeds onto this compressed soil layer. This mechanism enhances the seed's ability to utilize moisture from the lower soil layer, ultimately increasing germination rates.

While shallower seeding depths with double-disc openers at high residue improved emergence times, reduced emergence percentages suggest a complex interplay between soil moisture, seed-to-soil contact, and residue cover. Investigating additional factors like soil moisture content and alternative furrow opener designs for high-residue conditions could provide valuable insights for improving emergence under these scenarios. Further studies exploring residue management techniques to optimize these factors are also warranted.

## 4. Conclusions

The interplay between residue cover and furrow opener configurations in no-tillage systems was investigated in this study. Plant spacing showed no significant variation, but variation coefficients indicated a more uniform distribution at lower residue cover percentages. Hoe-type furrow openers maintained a consistent seeding depth regardless of residue cover or disc coulter application. Double-disc furrow openers exhibited a reduced seeding depth at high residue cover (80–90%), potentially due to residue interference. When seeding in high-residue-cover conditions, it is recommended to use row cleaners in front of the double-disc-type furrow openers since they tend to differ in seeding depth from hoe-type furrow openers.

MET and PE were affected by residue cover, with higher MET values and lower PE observed in double-disc-type furrow openers at increased residue cover rates. Disc coulters influenced MET and PE, emphasizing the need for a balanced approach in their selection.

While reduced seeding depth impacted MET and PE in high-residue-cover ratios, excessively low depths pose risks to seed viability, especially in warmer regions. Implementing appropriate furrow opener types and residue management strategies is crucial for optimizing planting depth, emergence uniformity, and seedling establishment in no-tillage systems with varying residue cover densities. Our findings provide practical insights for optimizing residue cover and furrow opener configurations, enhancing the efficiency of no-tillage seeding practices.

**Author Contributions:** D.K., conceptualization, formal analysis, investigation, methodology, validation, writing—original draft preparation; E.J., methodology, supervision, writing—original draft preparation; E.Š., conceptualization, methodology, writing—review and editing, funding acquisition. All authors have read and agreed to the published version of the manuscript.

**Funding:** This research received no external funding.

**Data Availability Statement:** The data presented in this study are available on request from the corresponding author.

**Conflicts of Interest:** The authors declare no conflicts of interest.

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
