# Peer review of "The Effect of Furrow Opener and Disc Coulter Configurations on Seeding Performance under Different Residue Cover Densities"

_agriengineering, doi:10.3390/agriengineering6020073_

Round 1

Reviewer 1 Report

Comments and Suggestions for Authors

Furrow opener type and residue cover has essential influence on seeding performance of no-till seeders. This manuscript conducted a study to investigate the impact of residue cover  and furrow opener type on no-tillage maize seeding performance of plant spacing, seeding  depth, mean emergence time, and percent emergence. Results can help optimize residue management for no-till farming systems.

1. Under high residue cover, hou-type opener seeding depth is deeper, why.

2. The residue amount is controlled by sowing, does it reached the presupposed taget value? 

Author Response

Thank you very much for your valuable comments. Your comments are quite useful for improving the paper. We have improved the manuscript and made further changes accordingly. We hope these changes will meet with your approval. These changes are listed below in a red” color.

Furrow opener type and residue cover has essential influence on seeding performance of no-till seeders. This manuscript conducted a study to investigate the impact of residue cover and furrow opener type on no-tillage maize seeding performance of plant spacing, seeding depth, mean emergence time, and percent emergence. Results can help optimize residue management for no-till farming systems.

  1. Under high residue cover, hoe-type opener seeding depth is deeper, why.

Response: The reason for the hoe-type opener seeding depth being deeper lies in the direction of the perpendicular component of the soil resistance force when the furrow opener cuts the soil. As illustrated in Figure 4, the perpendicular component (FP) of the resultant soil resistance acting on the hoe-type furrow opener is downward, facilitating the furrow opener's penetration into the soil. In contrast, for double-disc type furrow openers, FP is upward, compelling the disc to emerge from the soil. Consequently, the hoe-type furrow opener is not influenced by residue cover density in terms of seeding depth due to its construction. This situation is explained in lines 272-291 of the revised version of the manuscript.

  1. The residue amount is controlled by sowing, does it reached the presupposed taget value? 

Response: Conservation tillage aims to ensure that at least 30% of the soil surface is covered with crop residue. The approximate percentage rates of residue in the plots where the tests were carried out were 40%, 55%, 80%, and 90%. Therefore, in the study, the effects of percent residue covers of 40%, 55%, 80%, and 90% on the seeding quality for different furrow openers (hoe and double disc types) and disc coulters (control, plan disc, and ripple disc) combinations were examined. The residue rates were over 30%, which is the minimum rate for conservation tillage. Therefore, the residue amount reached the presupposed target value for conservation tillage.

This situation is explained in lines 200-201 and 213-221 of the revised version of the manuscript.

Again, thank you very much for your careful review; we agree that it has brought us a lot of inspiration. We have revised the manuscript to make the article more rigorous. We hope these changes will meet with your approval. Thank you again for your valuable comments on improving the quality of our manuscript.

Reviewer 2 Report

Comments and Suggestions for Authors

1. Suggest adding the number of references.

2. Suggest writing the analysis content of section 3 in a hierarchical manner, rather than as a whole part.

3. The formula for calculating the coefficient of variation was not mentioned in the paper.

4. What is the basis for the experimental design scheme in Table 1? Has the spatial distribution uniformity of variable factor selection been considered in the experimental design of the paper? Why did the author not choose the response surface method for analysis?

5. The content of the analysis section is too single, it is recommended to supplement it.

Author Response

Thank you very much for your valuable comments. Your comments are quite useful to improve the paper. We have improved the manuscript and made further changes accordingly. We hope these changes will meet with your approval. These changes are listed below in a red” color.

  1. Suggest adding the number of references.

Response: The manuscript cites 27 references in total, indicated with numbers according to the guide for authors of the Journal of AgriEngineering (as marked in red). All were checked.

  1. Suggest writing the analysis content of section 3 in a hierarchical manner, rather than as a whole part.

Response: Section 3 is divided into 3 subsections in a hierarchical manner with the following subheadings.

3.1. Plant (seed) Distribution Uniformity in the Row (Line 232)

3.2. Seeding Depth Uniformity (Line 252)

3.3. Percent Emergence and Mean Emergence Times (Line 315)

  1. The formula for calculating the coefficient of variation was not mentioned in the paper.

Response: The formula for calculating the coefficient of variation is presented in lines 168-174

  1. What is the basis for the experimental design scheme in Table 1? Has the spatial distribution uniformity of variable factor selection been considered in the experimental design of the paper? Why did the author not choose the response surface method for analysis?

Response: While the Response Surface Method is a powerful tool for exploring response surfaces and optimizing process parameters, it was not selected for analysis in this study. Our research focus was on comparing the effects of specific factors (furrow opener and disc coulter configurations) rather than optimizing a response function across a continuous design space. ANOVA was deemed suitable for comparing means across categorical factors within our experimental design.

In our experimental design, we carefully considered ensuring the spatial distribution uniformity of variable factor selection. Randomization techniques were employed during the setup of the experiment to minimize any potential bias due to spatial effects. Each treatment combination of furrow opener and disc coulter configurations was randomly assigned within each replicate block to account for any potential spatial variability in the field. Additionally, the use of a split-split plot design allowed us to distribute the treatments systematically across the experimental units while maintaining the integrity of the experimental design. By implementing these measures, we aimed to reduce the impact of spatial variation and ensure the validity and reliability of our experimental results.

We appreciate the esteemed reviewer's suggestion for the Response Surface Method and will consider it for our future research on optimizing process parameters.

The texts between lines 187-193 are added to explain the spatial distribution uniformity of factors.

  1. The content of the analysis section is too single, it is recommended to supplement it.

Response: The analysis/discussion has been expanded. New texts have been added to the manuscript to expand the analyses/discussions of the results (lines 242-246, 249-251, 298-303, 369-375).

Thank you very much for your careful review again; we agree that it has brought us a lot of inspiration. We have revised the manuscript to make the article more rigorous. We hope these changes will meet with your approval. Thank you again for your valuable comments on improving the quality of our manuscript.

Reviewer 3 Report

Comments and Suggestions for Authors

The article titled The Effect of Furrow Opener and Disc Coulter Configurations on Seeding Performance under Different Residue Cover Densities is very interesting. The research conducted by the Authors is carried out in accordance with current requirements. The test results are supported by appropriate calculations. The results are well described. Congratulations to the Authors.

Author Response

The article titled The Effect of Furrow Opener and Disc Coulter Configurations on Seeding Performance under Different Residue Cover Densities is very interesting. The research conducted by the Authors is carried out in accordance with current requirements. The test results are supported by appropriate calculations. The results are well described. Congratulations to the Authors.

Thank you very much for your thoughtful and encouraging review of our manuscript. We are thrilled to hear that you found our research interesting and that it met the current requirements. Your congratulations mean a lot to us, and we are truly grateful for your positive feedback. Your insights and encouragement will undoubtedly strengthen our commitment to producing high-quality research.

Once again, thank you for your time and attention to our work.
